# State-of-Art of Cellular Therapy for Acute Leukemia

**DOI:** 10.3390/ijms22094590

**Published:** 2021-04-27

**Authors:** Jong-Bok Lee, Daniel Vasic, Hyeonjeong Kang, Karen Kai-Lin Fang, Li Zhang

**Affiliations:** 1Toronto General Hospital Research Institute, University Health Network, Toronto, ON M5G 1L7, Canada; jongbok.lee@uhnresearch.ca (J.-B.L.); dan.vasic@mail.utoronto.ca (D.V.); Heidi.kang@mail.utoronto.ca (H.K.); k.fang@mail.utoronto.ca (K.K.-L.F.); 2Department of Laboratory Medicine and Pathobiology, University of Toronto, Toronto, ON M5S 1A8, Canada; 3Department of Immunology, University of Toronto, Toronto, ON M5S 1A8, Canada

**Keywords:** acute leukemia, adoptive cellular therapy, CAR-T cell, NK cell, DNT cell

## Abstract

With recent clinical breakthroughs, immunotherapy has become the fourth pillar of cancer treatment. Particularly, immune cell-based therapies have been envisioned as a promising treatment option with curative potential for leukemia patients. Hence, an increasing number of preclinical and clinical studies focus on various approaches of immune cell-based therapy for treatment of acute leukemia (AL). However, the use of different immune cell lineages and subsets against different types of leukemia and patient disease statuses challenge the interpretation of the clinical applicability and outcome of immune cell-based therapies. This review aims to provide an overview on recent approaches using various immune cell-based therapies against acute B-, T-, and myeloid leukemias. Further, the apparent limitations observed and potential approaches to overcome these limitations are discussed.

## 1. Introduction

Acute leukemia (AL) is a form of blood cancer originating in the bone marrow (BM) [1,2,3]. Due to abnormal fast growth of white blood cells, patients often experience fatigue, bleeding, bruising, fever, and a compromised immune system. Based on the lineage origins, AL can be categorized as B cell acute lymphocytic leukemia (B-ALL), T cell acute lymphocytic leukemia (T-ALL), and acute myelogenous leukemia (AML). ALL is more commonly seen in children, whereas AML is a more common form of AL in elderly population. First-line of treatment for ALs includes chemotherapy and radiation. Despite initial effectiveness, many patients experience disease relapse, leading to overall low long-term survival [2,4,5]. Hematopoietic stem cell transplantation (HSCT) is a commonly used and potentially curative second line of treatment for high-risk leukemia patients [6,7,8]. The treatment efficacy of HSCT comes from donor stem cell-derived immune cells, which have provided the foundation for adoptive cellular therapy (ACT) in patients refractory to or relapsing after conventional treatments [6]. The use of ACT for leukemia patients has become increasingly common and achieved notable success in certain leukemia types, possibly due to the sharing of the same niche between malignant blast and immune cells. This review summarizes recent advances and hurdles in using different types of cell-based therapies in treating different ALs, as well as future directions for the field.

## 2. B Cell Acute Lymphocytic Leukemia

ALL accounts for almost a quarter of all cancers in patients under the age of 20 [9]. While most cases happen in younger people, adult ALL patients are more challenging to treat and show worse prognosis [2]. Most ALL cases develop from B-cell lymphoid precursor cells, with B-ALL making up about 75% of all ALL cases [2]. Current standard-of-care includes various chemotherapy agents and HSCT for eligible patients [2]. However, lasting remissions are uncommon, and the rate of long-term survival for relapsed or refractory patients can be extremely low [1,2,10,11]. In recent years, novel autologous chimeric antigen receptor-modified T cell (CAR-T) therapies targeting the B-cell lineage antigen, CD19, emerged as a new hope for B-ALL patients with limited treatment options [10]. Some of the key clinical studies using CAR-T therapy for B-ALL treatment are listed in Table 1. Clinical trials treating relapsed and refractory B-ALL patients with anti-CD19 CAR-Ts (CAR19-Ts) reported remarkable results with remission rates between 68–93% [10,11]. However, several CAR-T-mediated adverse events were also reported, such as B-cell aplasia, cytokine release syndrome (CRS), and neurotoxicity [12,13]. As CAR19-Ts are directed against CD19, they also lyse non-leukemic B cells, causing B-cell aplasia, especially seen in patients that respond well to the treatment [11], leading to hypogammaglobulinemia [10,14]. However, such on-target off-tumor effects are treatable by intravenous administration of immunoglobulin to prevent any severe infections [11]. CRS is caused by high levels of cytokine secretion by CAR-T cells during cancer lysis, triggering monocytes to release a cascade of pro-inflammatory cytokines such as IL-6 and IL-1 [12]. With current CAR19-T treatments for B-ALL, over 75% of patients will normally develop some level of a CRS response [15]. When severe CRS develops, it can lead to multiorgan dysfunction and death [15]. Currently, the most commonly used treatments to mitigate severe CRS include tocilizumab, an antagonist of the IL-6 receptor, and siltuximab, an antibody against IL-6, with great efficacy in controlling the major complications [15]. While the root cause of CAR-T-mediated neurotoxicity is less well-understood, it seems to be related to the hinderance of the blood-brain-barrier caused by the increase in various inflammatory cytokines and other factors in blood after CAR-T treatment [12,16]. Incidence and severity of neurotoxicity varies amongst different studies, but commonly coincides with CRS [15]. If severe, neurotoxicity requires careful monitoring in the intensive care unit with the use of drugs such as levetiracetam, to mitigate the effects of the central nervous system toxicities [17]. Commonly observed treatment-associated toxicities experienced by B-ALL patients from CAR-T cell therapy are summarized in Table 2. Nevertheless, CAR-T therapy for B-ALL patients with no other effective treatment options outweighs these potentially severe side-effects, resulting in the FDA approval of autologous CAR19-T therapy for relapsed and refractory B-ALL [10,18].

### 2.1. Limitations of Autologous CAR-T Therapy

While autologous CAR19-T therapies are extremely promising, this individualized treatment still faces many challenges such as high treatment costs, lengthy vein-to-vein time due to long production processes, and potential manufacturing failures, challenging patients’ access to a life-saving treatment. Current cost estimates for CAR-T treatments are about $450,000, with the bulk of that coming from the cost of CAR-T product manufacturing (~$373,000) [23,24].This cost can be further increased with other CAR-T therapy-related interventions such as pre-conditioning lymphodepletion, adverse event management, and such high treatment-associated costs make CAR-T therapy less accessible to patients-in-need and pose a significant financial burden for the healthcare system to establish it as a standard-of-care for B-ALL.

CAR19-T studies reported about 10% of manufacturing failures due to the lack of abundance and quality of T cells from patients [24,25]. Also, about 13% of the patients fail to receive CAR-T cell treatment due to disease progression during the manufacturing process. Another hurdle in current CAR-T cell therapy is the quality of the final CAR-T product. As CAR-T cells are produced from each patient’s apheresis samples, not only has there been cancer cell contamination in the product, which led to CAR-expressing leukemic cell relapse, but also there is variability between the CAR-T products given to each patient, contributing to variable response rates [24]. Investigators observed that T cell exhaustion and memory status of the infused CAR-Ts varies between the responders and non-responders, clearly indicating that the quality and characteristics of the final CAR product is important for the success of the treatment [26].

Furthermore, despite the remarkable initial response rates, long-term outcomes of CAR-T-treated patients need to be improved [10]. For example, in one trial 83% of CAR-T treated patients achieved complete remission (CR). However, the median event-free survival was only ~6 months with a median overall survival of just over one year. Based on data extrapolation, the expected 5-year survival for these patients is 20–30%. One of the major reasons for disease relapse is downregulation of the target antigen on leukemic cells [21,27]. In six different clinical trials using CAR19-T for the treatment of B-ALL, 16–75% of the relapses were caused by CD19-negative B-ALL cells, seemingly more common in pediatric patients [28]. Therefore, despite the breakthrough, many investigators are trying to overcome the apparent hurdles of the current form of autologous CAR19-T therapy.

### 2.2. Allogeneic CAR-T Therapy for B-ALL

One approach to overcome some of the challenges of autologous CAR-T therapy is the use of healthy donor-derived allogeneic T cells as CAR vehicles [29,30]. This can reduce the waiting-time for patients to receive treatments, the product manufacturing cost, and product variability. Unlike the autologous manufacturing of CAR-T cells, healthy and functional donor T cells can be pre-screened and manufactured for CAR-T product, which can be cryopreserved or directly infused into the B-ALL patient as soon as required [29]. Also, an allogeneic model for CAR-T manufacturing circumvents the worry of a manufacturing failure in a time sensitive environment, and provides options for multiple dosing, something that is more challenging in an autologous setting [31]. In addition, the allogeneic platform can help to standardize the manufacturing processes, allowing for scaled up production and therefore significantly reducing the aggregate costs and accessibility of the treatment. One study showed that the aggregated cost of goods of autologous manufacturing was over 21 times more expensive per dose than allogeneic manufacturing processes, compared in two different facilities with identical capabilities [31]. The pros and cons of autologous versus allogeneic CAR-T therapy are summarized in Table 3.

Despite these advantages, adoptive transfer of donor T cells can cause serious toxicities such as graft-versus-host disease (GvHD) and mediate hindered anti-leukemic activity due to host-versus-graft (HvG) rejection [29]. To circumvent these issues, investigators have employed strategies such as endogenous T cell receptors (TCRs) and human leukocyte antigens (HLAs) knockouts, the main mediators of GvHD and HvG, respectively. In addition, other groups have knocked out CD52 from CAR-Ts to allow selective depletion of host effector T cells via anti-CD52 treatment [30,32]. Allogeneic CAR-Ts have been an area of increasing pre-clinical study in recent years, with a great deal of interest and investment from many institutions and biotechnology companies around the world. There are currently several ongoing early phase clinical trials, such as NCT02746952, NCT02808442, NCT03166878, NCT03398967, NCT0366000, and NCT03229876, utilizing genetically modified allogeneic CAR-Ts to treat B-ALL [29,33]. Results from two Phase I clinical trials (NCT02746952, NCT02808442) using the allogenic CAR-T product, UCART19, to treat aggressive pediatric and adult B-ALL have been released [19]. The UCART19 product was manufactured by lentiviral transduction of donor T cells with an anti-CD19 CAR construct, including genetic knockouts of the endogenous TCR and CD52. In terms of adverse events, 91% of patients developed CRS, 38% developed neurotoxicity, and 10% of patients experienced acute GvHD, despite efforts to mitigate its development. Despite 67% of UCART19-treated patients achieving CR, only 27% of the patients showed no evidence of disease progression with an overall survival of 55% at six months post treatment [19]. The promising results from these early studies demonstrate the feasibility of using allogeneic CAR-Ts as well as the need for further improvement in overall safety and long-term efficacy.

### 2.3. Alternate Vehicles for CAR Therapy against B-ALL

Additional genetic modifications and procedures required for allogeneic CAR-T therapy to prevent adverse events and improve therapeutic efficacy further complexify an already complicated as well as expensive manufacturing process. Also, unless there is complete knockout efficiency, issues such as GvHD and HvG rejection can still counteract the overall benefits of the treatment [29]. For all these reasons, the field is also investigating alternate immune cell vehicles beyond conventional αβ T cells to deliver highly effective, safe, and accessible CAR therapies. Natural killer cells (NKs), γδ-T cells, NK T cells, cytokine induced killer cells (CIKs), and double negative T cells (DNTs) are some of the vehicles that have been investigated as potential candidates [8,34,35,36]. Another potential benefit of using these cells as CAR-vehicles are their endogenous cytotoxic capabilities, aiding both overall efficacy as well as the potential prevention of antigen-escape relapse [37].

The relative safety and endogenous cytotoxic capabilities of CAR-NKs made them one of the first attractive candidates to deliver allogeneic CAR therapy [38,39]. There are three on-going and three planned clinical trials on anti-CD19 CAR-NK therapy, where the results of a Phase I trial using anti-CD19 CAR-NKs to treat B-ALL are expected soon. For CAR-NK therapy, in vivo persistence was a concern as it has been a major issue reported in other clinical trials utilizing NKs [20,39]. Recently, CAR constructs engineered to induce IL-15 secretion have shown great promise in increasing CAR-NK longevity in both mice and in the clinic [20]. In a phase I clinical trial, 73% of relapsed/refractory lymphoma and chronic lymphocytic leukemia (CLL) patients responded to anti-CD19 CAR-NKs with 64% of patients achieving CR [20]. In this study, CAR-NKs were expanded ex vivo from donor cord blood and 11 patients received either 0.1, 1 or 10 million cells per kilogram. The infused cells were detectable in all patients for at least 12 months without reported cases of GvHD, CRS, or neurotoxicity, supporting the safety of CAR-NK therapy. Notably, 3 of 11 patients experienced disease relapse and progression even with the presence of persisting CAR-NKs, suggesting other potential evasion mechanisms that need to be further explored. It is important to note that several patients that responded to the treatment also received post-remission treatments such as rituximab or venetoclax, making it difficult to evaluate the durability of response of the study. While this clinical trial is specifically treating lymphoma and CLL patients, the efficacy of CD19-targetting by CAR-NKs is expected to be translatable to B-ALL as was seen with CAR19-T cell therapies.

γδ-T cells are a T cell subset known to play important roles in cancer immunity and share similar characteristics and functions to innate immune cells [37,40]. Unlike conventional αβ-T cells, γδ-T cells function in an HLA-independent manner through engagement of the γδ-TCR as well as other various activating receptors such as NKG2D and DNAM-1 [40]. γδ-T cells are usually sourced from the peripheral blood and expanded ex vivo using phospho-antigens such as zoledronate, preferentially enriching for γ_9_δ_2_T cell subset. The lower cytokine secreting capacity of γδ-T cells as well as the lack of the αβ-TCR make γδ-T cells treatment relatively tolerable to patients [41,42]. However, issues with limited in vivo persistence and lack of standardized large-scale ex vivo expansion protocols remain [43]. A recent pre-clinical study investigated the feasibility of CAR19-transduction on zoledronate-expanded γδ-T cells and showed that CAR19-γδ-T cells target both CD19 positive and negative leukemic cells [42]. While some studies have demonstrated the safety, feasibility, and limited efficacy of non-modified γδ-T cells, the clinical applicability of CAR-γδ-T cells for B-ALL remains to be tested.

Our group is currently working on the potential utility of healthy donor-derived CD3^+^CD4^-^CD8^-^ DNTs as a vehicle for CARs. DNTs are a rare subset of mature T cells that can be expanded ex vivo from the peripheral blood of healthy donors [8,44,45]. It was demonstrated that DNTs fulfill the requirements of an off-the-shelf ACT including donor-independent function, cryo-preservability, resistance to HvG rejection, and no observed off-tumor toxicity, thus increasing their clinical utility at a lower cost [34]. DNTs can target an array of cancer types, but the level of cytotoxicity observed was modest against B cell malignancies [34]. However, our unpublished data indicate that transducing DNTs with a CAR19 significantly increased their ability to target B-ALL without observed toxicity in xenograft models. Hence, incorporating CAR19 or other CARs targeting B cell lineage markers (see below) onto DNTs may provide an alternate treatment that is safe, accessible, and a potent off-the-shelf-ACT against CD19^+^ B-ALL.

### 2.4. Recent Developments in CAR-T Therapy for B-ALL

While there has been a great deal of innovation surrounding the internal signaling domains of the CAR construct (discussed below), investigators have also been elucidating the possibilities for CAR with multiple Fab fragments to target B-ALL. CARs that target other B-cell lineage antigens such as CD20 and CD22 have been incorporated in the hope of combatting both tumor heterogeneity as well as antigen escape with dual targeting CARs such as the anti-CD19/CD22 CAR [21,27]. These dual, and in some studies triple, targeting CARs have shown superior results in pre-clinical studies than single antigen-targeting CAR-Ts, leading to multiple clinical trials [21,27]. Encouraging preliminary results from a phase I clinical trial using anti-CD19/CD22 CAR-Ts were reported, with 8/11 B-ALL patients achieving CR without evidence of antigen-negative escape, despite the short follow up of 60 days [21]. On the other hand, a couple of trials still reported B-ALL relapse despite the utilization of a dual targeting system [21,22].

Immune checkpoint inhibitors, including programmed cell death protein 1 (PD-1) blockade, have been well documented as a cornerstone of cancer immunotherapy. While most of its clinical benefits were observed in solid tumors, the ligands for PD-1 are expressed on a variety of hematological malignancies as well. Hence, approaches to combine CAR-T therapy with immune checkpoint blockade have been studied [46]. One pre-clinical study demonstrated that the expression of ligands for PD-1 on a B cell tumor model rendered the CAR-T non-functional, and that knockout PD-1 on CAR-Ts improved clearance of PD-ligand expressing tumor cells compared to the standard CAR-T [47]. However, another pre-clinical study reports that PD-1 knockout may actually harm CAR-T function in the long run by impairing their proliferative activity and, therefore, the durability of anti-leukemic response [48]. Currently, multiple clinical trials using PD-1 knockout CAR-Ts are in the recruitment phase [46]. The results from the aforementioned clinical trials will provide insights on the true potential of these combination style therapies.

Recent advancements in strategies to improve the safety profile of CAR-T treatments for B-ALL have also been made [49,50]. Many of the recent CAR constructs now include suicide switch genes, which allow for the inducible depletion of CAR-Ts from patients in the event of severe adverse events. The initial form of suicide genes was a thymidine kinase system which inhibits the CAR-T DNA replication through a drug, ganciclovir [49]. Due to the slower rate of CAR-T depletion using this method, more recent suicide switches rely on inducible death molecules such as Fas or caspase 9 [49]. In one study, 90% of the CAR-T cells bearing the inducible caspase 9 suicide gene died in less than an hour upon activation of the gene [51]. Alternative approaches to enhancing CAR-T safety include the use of an inducible CAR system, which induces the expression of CAR on T cells with a particular ‘on’ switch. In one study, a specific range of wavelengths of light induces the dimerization of a transcriptional activator that transcribes the CAR gene [52]. This group demonstrated the light controlled expression of the CAR on the surface of cells, as well as their efficacy both in vitro and in a mouse model after light stimulation [52]. However, while methods like this may be effective for small rodents or skin cancer or organs where certain wavelengths of light can easily penetrate, it remains to be seen whether innovative strategies like this can be feasibly translated into treating more systemic circulatory hematological cancers such as B-ALL. Similar approaches of on/off induction such as pharmacological, magnetic, or ultrasound-based methods are under investigation [53].

Collectively, advancements in cell therapy technology will help with improving the efficacy, durability, and accessibility of the treatment while reducing the treatment-associated toxicities to improve the overall outcomes for any patient diagnosed with B-ALL.

## 3. T Cell Acute Lymphocytic Leukemia

T-ALL is an aggressive hematological cancer of the T cell lineage that accounts for 15% to 25% of newly diagnosed pediatric and adult ALL cases [54]. Currently, the first-line treatment of T-ALL consists of chemotherapy with or without cranial radiation therapy [55]. Despite high initial response rates, 20% of pediatric and 40% of adult patients relapse within two years of diagnosis, and there are limited second-line treatment options [54]. Although allogeneic HSCT exists as a curative treatment for T-ALL, it requires remission reinduction and has risks of infections, GvHD, and relapse [55]. With breakthrough success of CAR-Ts for B-cell malignancies, researchers are investigating the use of CAR-T therapy as a potential treatment option for patients with T cell malignancies. In preclinical studies, various T cell-lineage antigens, including CD3, CD4, CD5, and CD7, have been examined as potentially targetable antigens using CAR. T cells from healthy donors transduced with CARs designed to target these antigens have shown potent cytotoxicity against T-ALL cell lines and primary tumors in vitro and in xenograft models [56,57,58,59,60,61]. Multiple constructs and generations of CARs have been studied to optimize the efficacy of CAR-Ts against T-ALL [56,62,63,64]. Presently, there are ongoing phase I clinical trials targeting T-ALL using CD4, CD5, and CD7- targeting CAR-Ts, where most are in the recruitment stage [65].

However, unique challenges associated with CAR-based immunotherapy for T-ALL have been observed. Below, the major challenges include the selection of targetable antigens, isolating a sufficient number of non-leukemic T cells, fratricide during manufacturing, and prolonged T cell aplasia (Figure 1), and ways to overcome them are discussed.

### Challenges of CAR-T Therapy for T-ALL

The first challenge lies in the selection of targetable antigens. CAR-T therapy for B cell malignancies often target B-cell lineage specific markers, such as CD19 and CD20 [10,27]. However, targeting T cell-lineage specific markers, such as CD1a, CD2, CD3, CD4, CD5, CD7, and CD8 can result in the killing of CAR-Ts themselves, a phenomenon called ‘fratricide’, during manufacturing due to expression of T-lineage markers on normal T cells. Indeed, fratricide has been observed in anti-CD5 and anti-CD7 CAR-T cells in preclinical studies [56,57,58]. Further, several of these antigens, including CD1a, CD2, and CD7 are also expressed on other cell lineages such as NKs and dendritic cells, creating the risk for on-target off-tumor toxicity.

In order to target T cell-linage antigens without fratricide, gene editing technology is used to eliminate the expression of the targeted T cell lineage markers from CAR-carriers prior to CAR-transduction. To develop CD3-specific CAR-Ts, Rasaiyaah et al. used transcription activator-like effector nucleases (TALEN)-mediated editing to disrupt endogenous TCRαβ/CD3 complex on effector T cells, followed by transduction with an anti-CD3 CAR [59]. Similarly, Gomes-Silva et al. used clustered regulatory interspaced short palindromic repeats (CRISPR)/Cas9 to disrupt CD7 prior to transduction with an anti-CD7 CAR [56]. In both studies, genetically modified T cells can express CARs without fratricide and mediate potent antitumor activity against T-ALL in vitro and in xenograft models [56,59]. The use of gene editing technology allows for efficient manipulation of CAR-Ts and versatility in the selection of antigen. However, there are concerns regarding off-target effects that can happen in gene editing, which may have unexpected consequences on CAR-T activity [66,67]. Also, such additional genetic modifications can lower the yield of CAR-T product. Manipulation of the CD3 complex, for example, negatively affects the proliferation of T cells, which potentially reduces the clinical availability of the treatment [59].

Due to the safety concerns of genetic editing, alternative methods to block surface antigen expression and prevent fratricide have been described. For example, Png et al. constructed a protein expression blocker (PEBL) based on an anti-CD7 single-chain variable fragment (scFv) coupled with an endoplasmic reticulum (ER)/Golgi-retention motif [57]. With transduction of the anti-CD7 PEBL, any newly synthesized CD7 remains in the ER or Golgi, which abrogates the surface CD7 expression. The authors show that PEBL transduction can effectively prevent fratricide without impairing CAR-T proliferation and cytotoxic function [57]. Unlike gene knockouts, PEBL transduction can be done using the same viral vector carrying the CAR, through either two sequential transductions or one single transduction using a bicistronic vector. However, it is important to consider the transduction efficiency, which can affect the yield of the final CAR-T product.

In another study, Ma et al. describe a strategy that redirects only CD8^+^ T cells to target CD4 -expressing T-ALL, which is about 50% of T-ALL cases [60]. In the study, CD8^+^ T cells that are modified to express anti-CD4 CAR show significant antitumor activity against CD4^+^ T-ALL in vitro and in xenograft models. The key strength of this approach is no need of additional genetic modifications. However, since CD4^+^ T cells are involved in the secondary expansion, memory formation for CD8^+^ T cells and enhancing the anti-leukemic effect of CD8^+^ T cells, the absence of CD4^+^ helper T cells in the CAR-T product may reduce the efficacy of such therapy [68,69].

In a study by Mamonkin et al., the degree of fratricide is reduced by minimizing the surface expression of CARs in vitro and inducing the CAR expression in vivo using drug-inducible CAR-system [58]. Anti-CD5 CAR constructs are coupled with the Tet-Off inducible expression system. CAR expression on transduced T cells is repressed in the presence of doxycycline in vitro. The removal of doxycycline restores CAR expression and antitumor activity of CAR-Ts. This strategy can be combined with different CAR constructs to provide a versatile way to expand CAR-Ts to sufficient numbers without fratricide during manufacture. However, with this approach, it may be important to consider the potential issues of limited CAR-T persistence and cancer relapse due to fratricide of CAR-Ts in vivo.

The second major hurdle of producing autologous CAR-T is the risk for lymphoblast contamination in autologous T cell products from T-ALL patients. As T-ALL patients have an abnormal T cell compartment, it is technically challenging to isolate sufficient number of healthy autologous T cells necessary for CAR-T cell therapy. Further, due to the lack of specific cell surface markers between normal and T-ALL cells, isolation of highly pure normal T cells from T-ALL patients can be more challenging than doing so from patients with other malignancies. In addition, there is the risk of transducing malignant T cells with CARs and re-infusing them into the patient. CAR-expressing cancer cells can mask the CAR-antigen and prevent recognition by CAR-Ts, leading to CAR-T resistant cancer [70]. Hence, more stringent purification steps are required for isolation of healthy autologous T cells prior to CAR-transduction.

The challenge of isolating healthy autologous T cells has steered studies toward the use of allogeneic T cells as a vehicle for CARs. To mitigate the risk of GvHD associated with the use of allogeneic T cells, Cooper et al. use CRISPR/Cas9 to disrupt the T cell receptor alpha chain (TRAC) and CD7 [61]. TRAC deletion prevents GvHD by blocking T cell receptor-mediated signaling, while CD7 deletion allows for anti-CD7 CAR-transduction without fratricide. These modified T cells demonstrate effective killing against T-ALL cell lines and primary T-ALL in vitro and in xenograft models without the induction of GvHD.

In addition, prolonged T cell aplasia is another potential risk of using CAR-T cells to target T-ALL. Upon engraftment, CAR-Ts can experience expansion rates of nearly 10^4^-fold in patients and result in adverse effects, such as cytokine storm or tumor lysis syndrome. Moreover, CAR-Ts can persist as memory cells, promoting long-term target depletion, as observed in CAR19-T therapy for B cell malignancies. Although B cell aplasia is shown to be clinically manageable, the effect of prolonged T cell aplasia is largely unexplored and requires further investigations.

An emerging approach to circumvent the challenge is by using NKs as a vehicle for CARs to treat T-ALL patients. As NKs lack expression of certain antigens found on T lymphoblasts, this approach can also reduce the risk for fratricide without requiring multiple genetic modifications. A human NK cell line, NK-92, is transduced with an anti-CD3, anti-CD4, anti-CD5, or anti-CD7 CAR [71,72,73,74]. All of the modified NK-92 cells exhibit robust antitumor activity against a variety of T cell leukemia and lymphoma cell lines and demonstrate significant reduction of tumor burden and prolonged survival in xenograft models of T-ALL. However, while NK cells show efficacy as an alternative vehicle for CARs, CAR-modified NK-92 cells are short lived and not detected in blood, BM, and spleen 17 days beyond last injection [73]. Thus, multiple booster doses of CAR-NKs were administered to reduce tumor burden in xenograft models [71,72,73]. However, CAR-NKs fail to completely eradicate established tumor cells, ultimately leading to disease relapse [73]. Conversely, some studies point out a potential benefit of NKs. Since NKs are exhausted shortly after tumor cell lysis, this may preclude the need for an inducible safety switch to prevent long-term T cell aplasia, as may be the case for CAR-Ts [75]. Given the unique challenges described above, to date, no CAR therapies have been explored in targeting T-ALL patients. The apparent challenges of CAR-T therapy for T-ALL treatment and potential approaches to mitigate them are summarized in Table 4.

## 4. Acute Myeloid Leukemia

AML is a form of hematological cancer characterized by clonal expansion of immature blasts of myeloid origin in BM [3,76], resulting in impaired hematopoiesis and BM failure [3,77]. AML is the most common form of adult AL, responsible for ~80% of adult AL cases [78,79]. The bulk of the disease occurs in an older population, and with an increase in the average life expectancies of human populations across the world, the incidence of AML has been rising [3,77]. Despite extensive studies done on the disease in the last few decades, the overall outcome remains poor with 30–45% survival for patients who are 60 years of age or younger and 10–15% for patients that are older than 60 years old [3].

### 4.1. Allogeneic Hematopoietic Stem Cell Transplantation for AML

Allo-HSCT is the only consolidation treatment with sustained curative potentials for the majority of AML patients. Donor-derived immune cells mediate graft-versus-leukemia (GvL) activity targeting residual blasts that are not killed by initial chemotherapy to prevent disease relapse. For patients with poor-risk AML, clinical benefit of allo-HSCT is more apparent as the probability of relapse drops from 70–90% to 30–50% [80,81]. Further, patient survival can be further enhanced by donor-lymphocyte infusion (DLI) or secondary HSCT, particularly when given during CR. However, secondary HSCT and DLI are not a treatment option readily available for all patients in need. Also, their clinical efficacy when given to patients in relapse or refractory to the disease is relatively modest. Hence, other therapeutic approaches that utilizes the potency of immune cell-mediated therapy are active area of research, particularly for those with relapsed or refractory AML. Novel approaches using immune cell-based therapies are summarized in Table 5.

### 4.2. CAR-T Therapy for AML

As CD19-directed CAR-T therapy effectively treats B cell leukemia, similar approaches were taken by targeting a myeloid specific marker, CD33, expressed on 85–90% of AML blasts [85]. In the initial clinical trial one refractory AML patient was treated with CD33-CAR-Ts which caused a marked decrease in BM blast level within two weeks, but gradual increase in blast level was observed by three weeks post infusion [82]. More thorough clinical studies on the safety and efficacy of CD33-CAR-Ts are currently on-going. CLL-1 is another potential molecular target for CAR-T cell therapy as CLL-1 is expressed on AML including leukemia stem cells [85]. A case study by Zhang et al. reported that secondary AML patient treated with anti-CLL CAR-T therapy achieved minimal residual disease (MRD) negative CR that lasted for 10 months [83]. In another study, two patients treated with CLL-1 and CD33 dual targeting CAR-Ts achieved MRD-negative CR within three weeks [84,85]. Subsequently, patients received HSCT, thus, durability of CLL-1 and CD33 dual CAR T response was not evaluable. Similarly, anti-CD123 CAR-Ts have been shown to achieve CR in two out of four patients with relapsing or refractory AML, when given at a higher dose of 2 × 10^8^ cells [86]. While these results support the potency of CAR-T therapy for treatment of AML, further investigations are needed to evaluate the response rate and durability of CAR-T therapy in AML.

In contrast to B-cell aplasia caused by CAR19-T therapy, the leading concern of AML-targeting CAR-Ts is the irreversible hematotoxicity caused by on-target off-tumor activity that can occur in addition to the common toxicities of CAR-T therapy [90,91]. Majority of CAR designed to target AML can recognize normal cells including hematopoietic stem cells, and hence, are associated with risks of BM failure [92]. To mitigate the treatment associated toxicities while utilizing the potency of CAR-based therapies, multiple studies focused on identification of antigens selectively expressed on AML cell surface and identified spliced variants of Flt3, NOTCH, and CD44 [93,94,95]. These targets showed comparable levels of anti-leukemic activity and lower off-tumor toxicity towards hematopoietic stem cells (HSCs) in preclinical models, but their safety and efficacy profile in clinical settings are not yet determined [96,97]. Alternatively, others have investigated targeting the previously identified antigens, such as CD33, while adding safety measures to attenuate the treatment-associated toxicity. For example, the feasibility of using transient anti-CD33-CAR expression [98], addition of suicide genes [99], and use of CD33 knockout HSCs as a follow-up treatment to anti-CD33 CAR-T treatment [90] were tested preclinically. However, whether these approaches can successfully be translated into the clinic is unclear and requires further investigations.

### 4.3. NK Therapy for AML

AML is a highly heterogenous disease that is associated with a low mutation rate. Hence, identification of cancer-associated antigens or neoantigen to treat many AML patients can be more challenging than with other cancer types [8]. Therefore, immune cells that can recognize malignant cells in an antigen non-specific manner is an attractive option for AML treatment. NK cell is a cell type extensively studied to treat AML in antigen non-specific manner as it can detect malignant cells through the balance of activating signals from NK receptors (NKRs), such as killer activation receptors (KARs), and inhibitory signals from certain killer-cell immunoglobulin-like receptors (KIRs) [100]. Activating receptors largely recognizes stress-induced ligands and these receptors include NKG2D, DNAM-1, and natural cytotoxic receptors, which have been shown to play an important role in AML recognition. In contrast, inhibitory KIRs interact with major histocompatibility complex (MHC) class-I molecules, which can be downregulated by cancerous cells to evade T-cell mediated immunity [101].

Compared to T cells, NKs have less robust alloreactivity. Therefore, many have demonstrated the safety of NKs as an AML therapy in an allogeneic setting. Among seven reported clinical trials, a total of 129 AML patients received allogeneic NK treatment, and none reported severe treatment-associated toxicity, although some experienced adverse events likely linked to preconditioning regimens [102]. While it should be noted that different studies utilized different NK products including purified NKs, HLA-KIR mismatched NKs, activated NK cell line, NK-92, and cytokine-induced memory like NKs, some studies also demonstrate the efficacy of NKs as consolidation therapy given to patients in CR to prevent disease relapse. Curti et al. showed that 9/17 high-risk elderly patients in CR treated with HLA-KIR mismatched allogeneic NKs as consolidation therapy remained in CR [103]. Further, Rubnitz et al. showed that all 10 young AML patients in first CR treated with purified KIR-HLA mismatched NKs remained in CR for a median follow-up of 964 days with a predicted two-year event free survival of 100% [104,105]. However, in a follow up study, among the 21 pediatric AML patients in CR treated with same therapy, 38% of them experienced disease relapse and 14% died of the disease with a median follow up of 1698 days [106]. In this study, the group concluded that the NK infusion did not improve the clinical outcome compared to similar patient population without NK infusion.

In a relapse or refractory setting, Romee et al. demonstrated the potency of NK cell-based therapy in AML patients with active disease, where AML patients that received allogeneic cytokine-induced memory-like (CIML) NKs achieved clinical responses in five out of nine treated patients without the need for KIR-HLA mismatch [88]. In another study, NK-92 was used to treat seven refractory/relapsing AML patients, and a short-lived response was observed in three patients [87]. Also, the study supported the potential of using NK-92 as an “off-the-shelf” cellular therapy for AML patients, but at the same time, revealed short persistence of the infused cells likely due to the irradiation of cell products prior to infusion to avoid in vivo tumorigenesis [107].

Use of NKs as vehicle for CAR to target AML has been investigated in multiple studies, mainly in the preclinical stage. Tang et al. investigated the safety and efficacy of anti-CD33-CAR expressing NK92, in three patients with relapsing/refractory AML [108]. Anti-CD33 CAR-NK92 was irradiated prior to infusion to avoid tumorigenesis. While the study supported the safety of the response, clinical efficacy was not obtained in none of the three patients. Clinical trials assessing the application of CAR-NKs targeting other AML-associated antigens, such as CD123 and CD7, are currently ongoing [85,97].

Different studies demonstrate the safety of therapeutic approaches using NKs for AML patient treatment with some encouraging efficacy data. However, it is unclear why some patients responded better than others. Ununified methods of NK manufacturing challenge interpretation of varying outcomes from different studies. In addition to variabilities of NK cell products, AML cells are a highly heterogenous disease that can employee various escape mechanisms against NK cell-mediated immunity, such as upregulation of KIR inhibitory ligands or release of soluble ligands to NK activating receptors [109]. Hence, future investigation into the characteristics of NK cells used in each trial and how the immune escape mechanisms employed by AML cells affect the clinical outcome of AML patients receiving NK therapy may shed light on better AML patient stratification for NK therapy.

### 4.4. DNT Therapy for AML

Our group investigated the therapeutic utility and the underlying mechanism of ex vivo expanded allogeneic DNTs against AML. In preclinical models, DNTs effectively targeted approximately 75% of the primary AML samples tested, including those obtained from refractory or relapsed AML patients [44]. DNTs effectively reduced leukemia blast engraftment in patient-derived xenograft models. Notably, allogeneic DNTs did not attack normal cells and tissues nor affect engraftment and differentiation of normal allogeneic HSC in various preclinical models [44]. Mechanistically, DNTs targeted AML cells in an NKG2D and DNAM-1 dependent manner, but were independent on T cell receptors. Notably, IFN-γ released by DNTs induced expression of NKG2D and DNAM-1 ligands on AML, further sensitizing them to DNT-mediated cytotoxicity. More recently, we have also identified FCGR1A and subunits of SAGA DUBm as AML susceptibility and resistance genes, respectively, to DNT cell-mediated cytotoxicity, using a CRISPR screen using a targeted sgRNA library [110].

Combining different therapeutic approaches that synergize can significantly improve the therapeutic outcome. To that end, it was observed that DNTs synergized with cytarabine and daunorubicin and Venetoclax and Azacytidine to evoke superior anti-leukemic activity in various AML preclinical models [111,112]. Cytarabine and daunorubicin are chemotherapeutic drugs commonly used first-line treatment for AML patients, and Venetoclax and Azacytidine are recently used as combination therapy with impressive clinical outcomes for treatment-naïve elderly patients [113,114]. Pretreating AML cells with the chemotherapeutic agents and azacytidine rendered AML cells more susceptible to DNTs including those otherwise resistant [111,112], and Venetoclax enhanced the potency of DNT-mediated cytotoxicity against AML by inducing ROS production [112]. Collectively, these results support the flexibility of DNT therapy used in combination with other anti-cancer therapies used for AML patient treatment.

Recently, our group completed a first-in-human phase I/IIa clinical trial using allogeneic DNTs to treat 12 patients with relapsing AML after allogeneic HSCT [89]. All 35 batches of cell products were successfully manufactured and infused as planned in this trial. None of the patients developed adverse events higher than grade 2. Notably, no patients showed any signs of GvHD or neurotoxicity, while all patients did develop grade I/II CRS, which were shortly resolved. The initial response was observed in 7 out 11 evaluable patients (6 achieved CR and 1 achieved partial remission (PR); 1 patient withdrew from the study for a personal reason). Five of the DNT treated patients remain alive and 4 in continuing CR 284 to 519 days after DNT-treatment. The predicted 1-year overall survival after DNT treatment was 43.6%. Although evaluation of the treatment efficacy remains a challenge due to small number of patients treated and heterogeneity amongst the patients, this trial supports the feasibility, safety, and potential efficacy of DNT therapy for the AML population with highly poor prognosis, and warrants a next phase clinical trial to evaluate the efficacy of DNTs in a larger patient cohorts.

## 5. Key Factors for the Success of Adoptive Cellular Therapy

With an increasing number of clinical studies on ACT for different forms of AL, the lack of durable treatment efficacy in the majority of treated patients becomes the major concern. One key factor that contributes to durable response is the persistence of infused cell products. A number of ACT clinical trials have commonly reported significant correlation between T cell persistence and durable clinical outcome [115,116,117,118,119,120]. Muller et al. reported that 52 complete responders of pediatric ALL patients from CAR19-T treatment showed persisting CAR-T cells in blood up to 24 months where three non-responders (NRs) showed rapid decay within the first month [121]. Also, Maude et al., reported that seven patients who achieved CR after CAR19-T infusion but had no detectable CAR-Ts within one-month experienced disease relapse between 6 weeks and 8.5 months post CAR-T infusion. In contrast, patients with sustained remission had detectable CAR-Ts for two years [122]. Similarly, among the 89 chemotherapy refractory AML patients treated with NK therapy, a higher density of persisting NKs in BM were seen in 55 patients who achieved remission (<5%, blasts in BM) whereas 34 patients still with the diseases (>5%, blasts in BM) showed lower levels of BM NKs, supporting a correlation between NK persistence and leukemia control [123].

### Characteristics of Infused Cells Affecting Their Persistence

Given the importance of cell persistence, different groups have attempted to identify factors associated with prolonged cell persistence. Firstly, proliferative capacity of infused cells in the first two weeks of infusion was shown to be correlated with their overall persistence in patients, and, therefore, a better patient outcome [121]. In ALL and CLL patients treated with CAR19-T therapy, the responders showed over 30-fold higher CAR19-T expansion in the peripheral blood than those in the NRs [121]. Similarly, B-ALL patients who received fludarabine and cyclophosphamide (Flu/Cy) lymphodepleting conditioning prior to CAR-T cell infusion had greater in vivo expansion and better clinical outcomes than patients received Cy alone as preconditioning [119]. These studies support the importance of proliferation of the infused cells, which can be promoted with more intensive preconditioning lymphodepletion to create a niche for infused cell products to expand. Although there is a significant association between expansion and persistence of infused cells, it should be noted that a great cell expansion can also correlate to the ACT-related toxicity such as CRS [121].

Polyfunctionality is another cell characteristic coupled with improved persistence. Polyfunctionality is measured by the ability of a single cell to mediate diverse immune reactions, such as the production of multiple effector cytokines and chemokines. Rossi et al. performed a single-cell multiplex cytokine analysis of CAR19-Ts and discovered a strong association between the polyfunctionality of the CAR product and expansion of CAR-Ts and improved clinical response in non-Hodgkin’s lymphoma patients [124].

Exhausted T cells express multiple inhibitory molecules, cytokine receptors, transcription factors, and effector molecules that are distinct from effector T cells [125,126]. T cell exhaustion defects its proliferative capacity as well as cytotoxicity against tumors. As such, high composition of exhausted T cells in the ACT product has been shown to be associated with poor persistence and poor clinical outcomes in multiple studies [26,127]. For example, Fraietta et al. showed a correlation between increased exhaustion transcriptomic profile of CAR-Ts and poorer response in CLL patients [26]. Similarly, Deng et al. also reported significantly higher expression levels of exhaustion associated genes, *LAG3* and *BATF*, within CAR19-Ts from diffuse large B-cell lymphoma (DLBCL) patients who had PR compared to those who achieved CR [127].

With an increasing amount of clinical data reported on the efficacy of ACT and their composition, the importance of memory subsets has become apparent. A significant correlation between transcriptomic signatures of memory T cells from the product and durable remission was seen, while non-responding patients were enriched in genes related to terminal differentiation and exhaustion of T cells [26,127]. In study conducted by Xu et al., 14 patients with relapsed/refractory B-cell malignancies received autologous CAR19-T therapy and showed a direct correlation between the expansion rate and the higher proportion of early memory subsets of the infused CD8^+^ T cell product, identified as CD45RA^+^CCR7^+^, which in turn, resulted in longer T cell persistence and better clinical response [128]. Collectively, these clinical observations led many scientists to investigate various strategies to improve T cell persistence by inducing T cell memory while inhibiting T cell exhaustion to enhance the overall efficacy of ACT.

## 6. Strategies to Improve Cell Persistence

Emerging pre-clinical studies investigated different strategies to improve cell persistence including alteration in manufacturing condition by introducing cytokines or pharmaceutical inhibitors and genetic modification and CAR construct modification (Figure 2).

### 6.1. Cytokines

Immune cell activation in the presence of IL-2 has been the most widely accepted ex vivo expansion method for various cell types including conventional T cells, NKs, and γδ-T cells. The effect of other γ-chain cytokines such as IL-4, IL-7, IL-21, and IL-15 on the persistence has been actively explored. It has been shown that ex vivo expansion of CD4^+^ and CD8^+^ T cells in the presence of IL-7 and IL-15 together can improve T cell persistence in vivo compared to those cultured in the presence of IL-2 by preserving T cells in an early differentiation state as well as preventing T cell exhaustion [129]. Another study demonstrated that ex vivo expansion of CAR19-T cells in the presence of IL-2 and IL-21 skewed T cells towards an early memory state with CD62L^+^CD28^+^ expression as well as with a transcriptional profile of naïve T cells [130]. These T cells showed superior efficacy against B-ALL in a xenograft model compared to CAR19-Ts expanded in the presence of IL-2 alone. To overcome limited in vivo persistence of NK-based therapy, NKs were expanded in the presence of IL-12, IL-15, and IL-18, generating CIML-NKs [88,102]. CIML-NKs exhibited significantly enhanced proliferation, expression of IL-2 receptor αβγ and increased IFN-γ production after restimulation, resulting in enhanced anti-leukemic function in preclinical settings. First clinical trial using IL-12, IL-15, and IL18 pre-activated CIML-NKs to treat refractory AML showed a great expansion of the infused cells in patient blood and BM and an encouraging overall response rate [88]. It is now widely accepted that use of certain cytokines can enhance the persistence of cellular products, and hence their anti-leukemic activity. However, the optimal cytokine composition during expansion that elicits the best performance of immune cell therapy remains to be standardized.

### 6.2. Small Molecule Inhibitors

The activation of certain signaling pathways can dictate the differentiation status of immune cells. As such, investigators have utilized small molecule inhibitors during cell manufacturing to finetune the signaling pathways to improve immune cell persistence. Among many inhibitors, the recent application of the mitogen-activated protein kinase kinase (MEK)1/2 inhibitor, phosphatidylinositol 3-kinase (PI3K)-δ inhibitor, Idelalisib, or the bromodomain and extra-terminal motif (BET) protein inhibitor, JQ1, to the ex vivo expansion of T cells, has shown to significantly enhance the persistence and therapeutic efficacy of ACT in various preclinical cancer models [131,132,133]. Expansion of T cells in the presence of these inhibitors commonly skewed them towards stem cell-like memory T cells that are associated with better self-renewability and proliferative capacity. Mechanistically, MEK1/2 inhibitor delays cell division and enhances mitochondrial fitness with increased fatty acid oxidation for energy generation [131]. Idelalisib attenuates activation signal of T cells from the engagement of TCR, co-stimulatory receptors, and cytokine receptors [132,134], and JQ1 suppresses the expression of transcription factor *BATF* in CD8^+^ T cells, a transcription factor involved in differentiation of CD8^+^ T cells into effector T cells [133]. However, the use of small molecule inhibitors to manipulate ACT production has not yet been translated into the clinic. Therefore, while encouraging, further validation of safety and efficacy in patients is needed.

### 6.3. Genetic Modification

Exhausted T cells have reduced effector function and expression of inhibitory receptors, thus having reduced persistence and efficacy in vivo. Therefore, different groups have attempted to reverse T cell exhaustion by means of genetic inhibition. For example, Stadtmauer et al. successfully overcame T cell exhaustion by knocking out PD-1 from CAR-Ts using CRISPR technology [135]. More recently, three separate studies identified a high mobility group (HMG)-box transcription factor, TOX, as a critical regulator for T cell exhaustion [136,137,138], where knocking out TOX prevented T cell exhaustion accompanied with lower expression of inhibitory receptors (*Pdcd1*, *Lag3*, *Ctla4*) and transcription factors (*Nr4a2*, *Ikzf3*, *Tox2*) associated with T cell exhaustion. Consistent with such findings, superior anti-tumor effects and persistence of TOX-deficient CAR19-T cells were observed [139]. Similarly, Wei et al. demonstrated that Regnase-1-depleted CD8^+^ T cells expressed higher levels of genes associated with naïve or memory cells and lower expression of TOX, and that these cells have markedly improved anti-leukemia activity and persistence in murine models [140]. Collectively, these studies demonstrate the applicability of genetic manipulation to target exhaustion or memory associated genes to improve the outcomes of ACT. However, translating these findings from bench to bedside still needs further development.

### 6.4. CAR Intracellular Signaling Domain

One of the heavily studied approaches to specifically enhance CAR-transduced immune cell persistence has been engineering of the composition and structure of the internal signaling domains of the CAR [141]. The earliest CAR constructs were composed of an extracellular antibody scFv antigen recognition domain linked to an intracellular CD3ζ activation domain derived from the TCR/CD3 complex. These first-generation CAR constructs were able to effectively target cancer cells, but due to the lack of proper T cell activation, showed extremely limited persistence and expansion in clinical trials [142]. In an effort to improve the overall function of CAR-Ts and to prevent T cell anergy after target recognition, the co-stimulatory domains, CD28 and 4-1BB, were added to the intracellular region of second-generation CARs. While CD28-based CAR-Ts have robust effector activity, strong signaling also led to a higher degree of activation-induced cell death and exhaustion [143]. On the other hand, downstream signaling of co-stimulatory domains from the tumor necrosis factor receptor (TNFR) family, 4-1BB, OX40 or CD27, initiates the activity of NF-κB through TNFR associated factor (TRAF) proteins, culminating in the upregulation of proteins involved in the regulation of cell cycle and survival such as MYC, cyclin D1, BCL-XL, and BCL-2 [144]. Hence, these CAR- T cells showed improved survival, persistence, mitochondrial biogenesis as well as oxidative metabolism. As expected, drastic differences have been observed in cell persistence, where CD28-containing CAR-Ts are often only detectable for a few months in patients, while 4-1BB-containing CAR-Ts are detected years after infusion [145]. However, despite these differences, the clinical efficacy of CAR-Ts bearing the two constructs have been comparable in B-ALL patients [146]. This suggests that CD28-based CAR-Ts remain to be effective against highly aggressive cells such as B-ALL, despite their reduced capacity to persist, possibly due to the robustness of their response.

Third generation CAR combines the intracellular domains of both CD28 and 4-1BB to yield a robust synergistic effect in preclinical and clinical studies [147]. With such observations that support the importance of intracellular signaling domains of CARs, novel fourth generation CAR constructs are beginning to make a bigger impact on the field. In addition to CD3ζ and CD28 or 4-1BB signaling, fourth generation CARs rely on inducible activation of a downstream transcription factor sensitive promoter [21,148]. The molecules induced by transgenic expression are usually various cytokines, including IL-12, IL-15, and IL-18, among others, which can help stimulate immune cell proliferation and survival, or have a modulatory role on the tumor microenvironment in an effort to minimize external pressures on the CAR-Ts. One group has developed a novel CAR construct that contains a JAK-STAT based co-stimulatory domain which induces cytokine signaling after target ligation, such as IL-21, which is known to facilitate T cell proliferation [149]. In pre-clinical studies, the JAK-STAT based CARs showed superior anti-cancer efficacy as well as persistence, relative to CD28 and 4-1BB based CAR-T cells, in both hematological and solid cancer models, and will be investigated in clinic studies.

## 7. Conclusions

Various approaches of adoptive immune cell therapy for different types of ALs revealed promising outcomes for preclinical models and patients, but also apparent hurdles for researchers to overcome (Figure 3). While the use of different immune cell subsets, expansion conditions, and genetic/pharmacological modifications have overcome some of the challenges, it also complicated the interpretation of the outcomes using ACT for AL treatment. Nevertheless, the importance of safe and targetable antigens and persistence of immune cells are common critical factors across different AL types. Also, outcomes from various clinical studies suggest that it is unlikely for one single ACT to be a safe and effective treatment option for all AL patients. Hence, active investigations on improving and developing new ACT approaches will enhance the efficacy and clinical application for AL. Also, the feasibility of combining ACT with other therapeutic approaches such as chemotherapy and checkpoint inhibitors may increase the overall outcomes in patients.

## Figures and Tables

**Figure 1 ijms-22-04590-f001:**
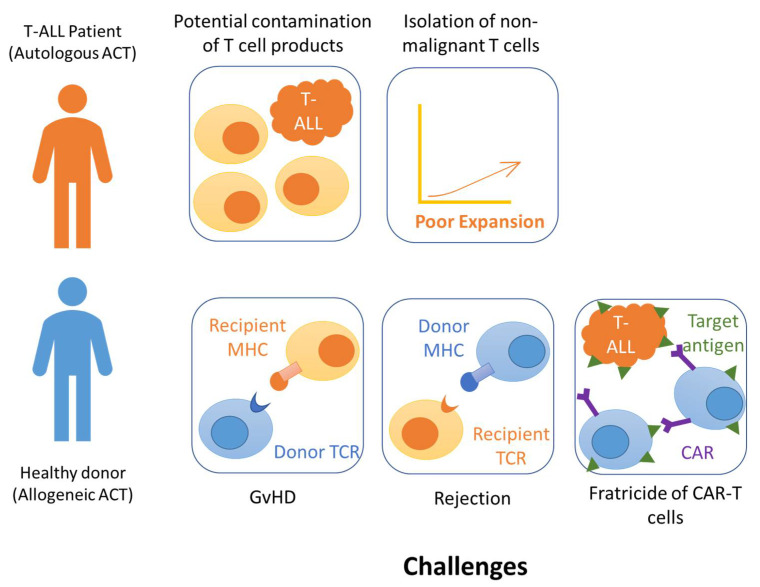
Challenges of CAR-T therapy for T-ALL.

**Figure 2 ijms-22-04590-f002:**
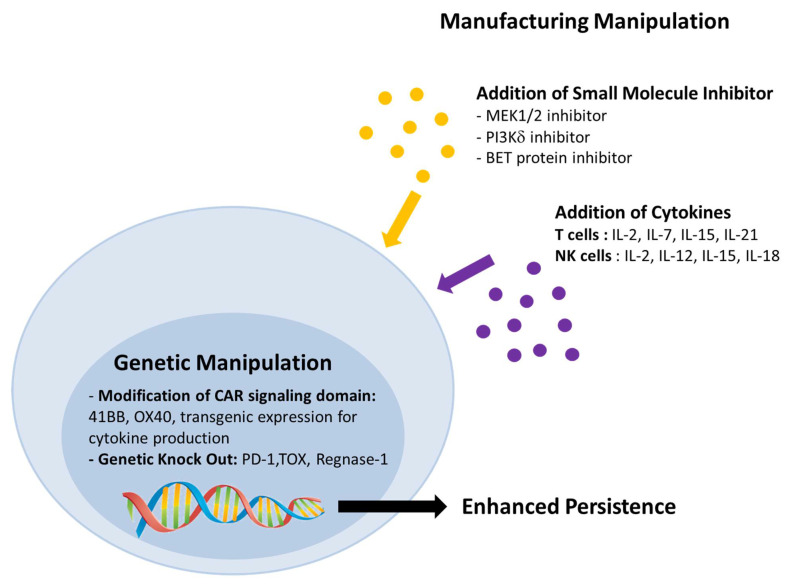
Various approaches to improve the persistence and therapeutic efficacy of adoptive cellular therapy.

**Figure 3 ijms-22-04590-f003:**
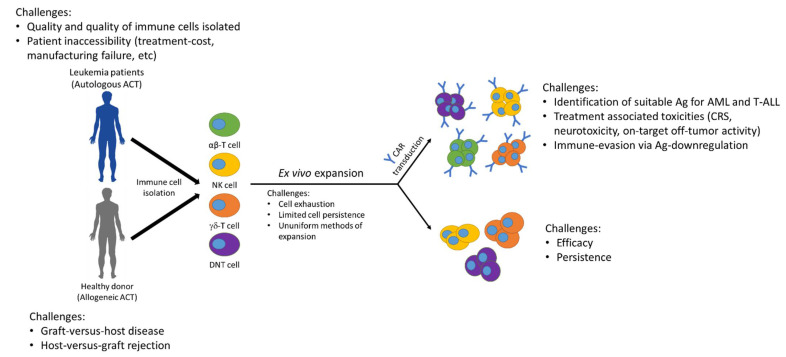
Current challenges using ACT to treat acute leukemia.

**Table 1 ijms-22-04590-t001:** Adoptive cellular therapies used to treat B-ALL in clinical trials.

Cell Type	Ag Targeted	Manufacturing Method	Pt #	Disease Status	Treatment Associated Toxicities	Disease Outcome	Notes	Clinical Trial #/ (Reference)
CAR-T cell	CD19	-Autologous T cell source -2nd gen CD28z CAR -Expansion with CD3/CD28 dynabeads	53	-Relapsed and refractory -Patients were all heavily pre-treated, third line treatment for 68% of patients	-CRS (n = 45, 85%) -Severe CRS (n = 14, 26%) -Neurotoxicity (n = 23, 43%)	-CR = 83% -Median EFS = 6.1 months -Median OS = 12.9 months	-Patients with high initial disease burden (>5% BM blasts) had higher incidence of AE and lower EFS and OS rates -CD19 negative relapse observed in 4 patients -Median time of T cell detectability = 14 days	NCT01044069 [10]
CAR-T cell	CD19	-Autologous T cell source -2nd gen 4-1BBz CAR -Expansion with CD3/CD28 dynabeads	75	-Relapsed and refractory, average 3 previous treatments -46% received previous alloHSCT	-CRS in 77% -Neurotoxicity in 40% -All responders had B cell aplasia	-ORR = 81% -EFS rate @ 6 and 12 months = 73% and 50% -OS rate @ 6 and 12 months = 90% and 76%	-96% of patients received pre-conditioning lymphodepletion -Median time to max expansion of T cells was 10 days	NCT02435849 [11]
CAR-T cell	CD19	-Allogeneic T cell source (batches from 3 different HD used) -2nd gen 4-1BBz CAR, rituximab mediated depletion epitope added for safety purposes -Genome-edited CAR-T cells, knocked out for TCRα chain and CD52 to prevent GvHD	21	-Relapsed and refractory, average 4 previous treatments -62% received previous alloHSCT -Median initial BM tumor burden = 8% blasts	-CRS (n = 19, 91%) -Neurotoxicity (n = 8, 38%) -GvHD (n = 2, 10%)	-CR = 67% -Median duration of response = 4.1 months -EFS rate @ 6 months = 27% -OS rate @ 6 months = 55%	-All patients received pre-conditioning lymphodepletion -3 patients saw CAR-T persistence beyond 42 days	NCT02808442 and NCT02746952 [19]
CAR-NK cell	CD19	-Allogeneic HLA-mismatched NK cell source (cord blood derived) -4th gen CAR that included inducible IL-15 secretion domain and caspase 9 safety switch -Expansion in presence of K562 feeder cells and IL-2	11	-Relapsed and refractory, all patients had 3 or more previous lines of therapy -All adult/elderly patients -Patients had lymphoma or CLL (CD19 targeting means results expected to be translatable to B-ALL patients)	-No AE -No GvHD -No increase in levels of inflammatory cytokines	-73% of patients had a response (n = 8) -CR = 64% (n = 7) -Response duration difficult to assess due to several patients receiving various post-remission therapies	-CAR-NK cells persisted at low levels for at least 12 months -All patients received pre-condition lymphodepletion -3 patients experienced disease relapse or progression	NCT03056339 [20]
CAR-T cell	CD19/CD22	-Autologous T cell source -Bispecific 2nd gen CAR, OX40 costim paired with CD19 and 4-1BB costim paired with CD22 -Semi-automated manufacturing process	10	-Relapsed and refractory, high-risk patients -All pediatric patients	-CRS (n = 9, 90%) -Neurotoxicity (n = 1, 10%) -Patients in ongoing CR maintain B cell aplasia -No patients required ICU admission	-CR = 100%, all MRD negative -3 patients relapsed within post treatment 1 year -Longer follow-up result updates still need to be reported	-All patients received pre-condition lymphodepletion -1 of the 3 relapses had CD19 negative/CD22 low relapse -180 days median persistence of CAR-T cells in blood	NCT03289455 [21,22]

AE = adverse events; Ag = Antigen; alloHSCT = allogeneic hematopoietic stem cell transplant; BM = bone marrow; CAR = chimeric antigen receptor; CR = complete remission; CRS = cytokine release syndrome; EFS = event free survival; GvHD = graft versus host disease; HD = healthy donor; HLA = human leukocyte antigen; ICU = intensive care unit; NK = natural killer; ORR = overall remission rate; OS = overall survival; Pt = patient; TCR = T cell receptor.

**Table 2 ijms-22-04590-t002:** Major CAR Treatment Related Side Effects for B-ALL Patients.

CAR Treatment Related Side Effects	Incidence	Complications	Treatment Strategies	References
B cell Aplasia	-Develops in most patients, especially when the treatment against the leukemic B cells is effective (patients in CR)	-Decreased B cell counts -Hypogammaglobulinemia -Severe infections if untreated	-IV administration of immunoglobulin (IG replacement therapy)	[10,11,12,13,14]
CRS	-Develops in many patients, correlates with strong CAR-T cell response, usually occurring within time of max cell expansion -Some grade of CRS can develop in over 75% of B-ALL patients	-High levels of pro-inflammatory cytokines such as IL-6 and IL-1 -Symptoms of cytokine storm -Very high fever, headache, fatigue, myalgia, organ dysfunction, etc.	-Common monitoring practices in the ICU until resolution (only if severe) -Analgesics, vasopressors, IV fluids, etc. -Tocilizumab (IL-6 receptor antagonist) -Siltuximab (IL-6 antibody) -IL-1 blockade -Anti-inflammatory drugs (methylprednisolone)	[12,13,15]
CNS Neurotoxicity	-Very wide range of incidence between studies (varies between no cases to majority of cases in a study) -Commonly coincides with strong CRS response, however can develop in absence of CRS as well	-More mild cases can include headaches, disorientation, memory and attention loss, impaired speech/writing, etc. -More severe cases can include seizures, encephalopathy, impaired motor and movement functions, coma, etc.	-Common monitoring practices in the ICU until resolution (only if severe) -Levetiracetam for certain neurological symptoms -Corticosteroids -Using strategies to reduce CRS response as contributing factor can help	[12,13,15,17]

B-ALL = B cell acute lymphocytic leukemia, CAR = chimeric antigen receptor, CNS = central nervous system, CR = complete remission, CRS = cytokine release syndrome, ICU = intensive care unit, IG = immunoglobulin, IV = intravenous.

**Table 3 ijms-22-04590-t003:** Pros and cons of autologous and allogeneic CAR-T therapy.

Immune Cell Source	Pros	Cons
Autologous	-Generally safer to infuse as there is no risk for other side effects beyond CAR mediated -No risk for HvG rejection, hence, cells may persist longer to induce more durable responses -Simpler manufacturing process, no need for additional genetic modifications -No need to search for other potential donors	-High treatment costs due to individualized nature, making one product for one patient -Poor quality and limited quantity of isolated immune cells -Potential manufacturing failures -Lengthy vein-to-vein time -Patients can experience disease progression or death while waiting for treatment -Not accessible to all patients
Allogeneic	-Off-the-shelf approach possible -Can use other immune cells other than T cells to potentially avoid GvHD -Genetic engineering techniques have advanced to help avoid treatment associated toxicities -More cost effective -Increased accessibility for all patients -Increased potential for multiple dosing if necessary -Can pre-screen donor cells to ensure that quantity and quality requirements can be met -Can potentially treat multiple patients with cells from one healthy donor -Increased speed between patient’s requirement for treatment and actual CAR-T cell infusion	-Potentially life-threatening GvHD can counteract any benefit of the treatment -HvG rejection can greatly limit cell persistence, hinder treatment efficacy and durability -Cost and complexity of techniques to avoid treatment associated toxicities might outweigh cost efficiency benefits of allogeneic approach

CAR = chimeric antigen receptor, GvHD = graft-versus-host disease, HvG = host-versus-graft.

**Table 4 ijms-22-04590-t004:** Challenges of CAR-T therapy against T-ALL and strategies to overcome the challenges.

Challenge	Strategy	Target Antigen Studied	Reference
Fratricide	Genetic editing of target antigen	CD3, CD7	[56,59]
	Protein expression blocker (PEBL)	CD7	[57]
	Tet-Off inducible expression system	CD5	[58]
	Using CD8^+^ T cells only	CD4	[60]
	Using CAR-NK cells	CD3, CD4, CD5, CD7	[71,72,73,74]
Potential contamination of autologous products	Allogeneic CAR-T cells with TCR editing	CD7	[61]
	Using CAR-NK cells	CD3, CD4, CD5, CD7	[71,72,73,74]
Prolonged T cell aplasia	Safety switch	CD4	[60]
	Using CAR-NK cells	CD3, CD4, CD5, CD7	[71,72,73,74]

**Table 5 ijms-22-04590-t005:** Adoptive cellular therapies used to treat AML patients in clinical trial.

Cell Type	Ag Targeted	Manufacturing Method	Pt. n	Disease Status	Treatment Associated Toxicities	Disease Outcome	Notes	Clinical Trial No./ (Reference)
CAR-T cells	CD33	-Autologous T cell source - Lenti-virally transduced - CD3ζ and 4-1BB intracellular domain -T cells were cultured in the presence of CD3 antibody, IL-2 and IFNγ for 11 days.	1	Relapsed	Grade 4 chills Exacerbation of existing pancytopenia	-BM blasts decreased from >50% to <6% @ 2wks -70% BM blasts @ 9wks	-CD33-CART persisted and retained anti-leukemic activity ex vivo -Blasts developed resistance to CD33-CART cells	NCT01864902 [82]
CAR T cell	CLL-1	-Autologous T cell source -Lenti-virally transduced - Fourth generation CLL- CAR containing CD3, CD28, and CD27 intracellular domain.	1	Secondary AML	Grade I/II CRS developed. Patient experienced hypotension	-MRD negative -Morphological CR	-CR retained up to 10 months. -A low CAR T-cell level persistence was detected 5 months after CAR T-cell injection	trial number not available. [83]
CAR T cell	CD33 and CLL-1	-Autologous T cell source - Anti-CLL1/CD33 CAR linked via a self-cleaving P2A peptide.	2	Refractory /relapsed	Pancytopenia	-MRD negative CR for both patients	-An alemtuzumab safety switch implemented to eliminate CAR T cells following tumor eradication.	NCT03795779 [84,85]
CAR T cell	CD123	-Autologous T cell source -Lenti-virally transduced - CD3 and CD28 intracellular domain	6	Refractory AML after allo-HSCT	Grade 1 or 2 CRS reported in most patients, but did not reach dose-limiting toxicities.	Morphological CR (n = 1) CR (n = 2) Partial response (n = 2)	-CR patients received secondary allo-HSCT	NCT02159495 [86]
NK-92	Unspecified	-Allogeneic NK source -Expanded in presence of IL-2 -Ex vivo expanded NK-92 cells were irradiated at 1,000 cGy prior to patinet infusion	7	Refractory/ Relapsed	No AE	-PD (n = 3) -Decreased BM blasts (n = 1) -Stable % BM blasts (n = 2) -NE (n = 1)	-2-6 infusions given per patient	NCT00900809 [87]
CIML NK cell	Unspecified	-Allogeneic NK cell source -Pre-activated with IL-12, IL-15, and IL-18. -Subsequently, cultured in the presence of IL-2	9	Refractory/ Relapsed	No AE	-CR/CRi (n = 4) -Morpohlogical CR (n = 1)	-13 patients enrollled. 4 were inevaluable due to manufacturing failure, death before evaluation of responses.	NCT01898793 [88]
DNT cell	Unspecified	-Allogeneic T cell source -CD3+ CD4- CD8- cells expanded in the presence of anti-CD3 antibody and IL-2	12	Relapsed after allo-HSCT	Grade I/II CRS	-CR/CRi (n = 6) -Partial remission (n = 1)	-1 patient withdrew from the study	ChiCTR-IPR-1900022795 [89]

AML = acute myeloid leukemia; NK = natural killer; AE = adverse events; CR = complete remission; NE = not evaluable; alloHSCT = allogeneic hematopoietic stem cell transplant; MRD = minimal residual disease; aNK = activated NK cells; PD = progressive disease.

## Data Availability

Not applicable.

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
