# Peer review of "State-of-Art of Cellular Therapy for Acute Leukemia"

_ijms, 2021, doi:10.3390/ijms22094590_

Round 1

Reviewer 1 Report

The authors review the existing data regarding the use of CAR T-cell therapy as a potential cure for ALL and AML. CAR T-cell therapy employs the ability of the immune system to identify and destroy malignant cells and when combined with chemotherapy and checkpoint inhibitors, can improve patient survival. The authors point out the key factors involved in the success of CAR T-cell therapy (the safe and targetable antigen, the persistence of infused cells products, their proliferative capacity and polyfunctionality etc.) and their apparent limitations.   As minor suggestions, I would recommend adding a table to summarize the most relevant studies analyzed for each acute leukaemia subtype, cells employed, targetable antigens, prior/post-chemotherapy used to enhance CAR T-cell efficacy, main adverse effects, overall survival and a cost-efficiency analysis. 

Reviewer 2 Report

This review is very interesting due to the large number of scientific events that are currently being addressed to treat and increase the life of patients with leukemia. The article gives a timely view of the different attempts to tackle this health scourge.

The authors could be instructed to make a summary table of the techniques and / or protocols that are giving the best results for the different types of leukemia that allow the clinician to have a global vision of the theme.

Reviewer 3 Report

In this review Lee and colleagues describe in an exhaustive manner the state-of-art of cell-based therapies in treating acute leukemia.

I think it is necessary to carry out changes in order to make the review more attractive for readers.

I think that a single figure is not enough to illustrate the main concepts reported in the manuscript. Furthermore, a summary figure inserted at the end of the introduction doesn’t make much sense. Inserting multiple images and tables would give more engaging to the article. For example:

  • Paragraph 2 (B cell Acute Lymphocytic Leukemia): insert a table summarizing CAR-T cell side effect and treatment options;
  • Paragraphs 2.1 and 2.2: insert a table o figure about pro-cons autologous and allogenic CAR-T cell therapy;
  • Paragraph 3: insert a figure to explain the potential outcomes of CAR-T cell therapy and “fratricide” phenomenon in T-ALL patients;
  • Paragraph 4.2: insert a figure about CAR-T therapy in AML;
  • Paragraph 6: figure to describe strategies to improve the efficacy of CAR-T cell therapy;
  • Summarize all cited adoptive cellular therapy clinical trials in a table.

Minor changes:

  • Line 21: enter the full name for acute leukemia as it is the first time it is mentioned in the text.
  • Line 261: rimove the full stop.
  • Lines 274 and 304: insert the reference.

Author Response

In this review Lee and colleagues describe in an exhaustive manner the state-of-art of cell-based therapies in treating acute leukemia.

I think it is necessary to carry out changes in order to make the review more attractive for readers.

We want to thank the reviewer for carefully reading our manuscript and providing an excellent suggestion to improve the quality of the review.

I think that a single figure is not enough to illustrate the main concepts reported in the manuscript. Furthermore, a summary figure inserted at the end of the introduction doesn’t make much sense.

We agree with the reviewer’s comment and have moved the summary figure to conclusion section as Figure 3.

Inserting multiple images and tables would give more engaging to the article. For example:

We have added following tables to address the reviewer’s suggestions:

Paragraph 2 (B cell Acute Lymphocytic Leukemia): insert a table summarizing CAR-T cell side effect and treatment options; (Table 1)

Paragraphs 2.1 and 2.2: insert a table o figure about pro-cons autologous and allogenic CAR-T cell therapy; (Table 2)

Paragraph 3: insert a figure to explain the potential outcomes of CAR-T cell therapy and “fratricide” phenomenon in T-ALL patients; (Figure 1)

Paragraph 4.2: insert a figure about CAR-T therapy in AML; (Table 3)

Paragraph 6: figure to describe strategies to improve the efficacy of CAR-T cell therapy; (Figure 2)

Summarize all cited adoptive cellular therapy clinical trials in a table. (Table 1 for B-ALL and Table 5 for AML)

Minor changes:

Line 21: enter the full name for acute leukemia as it is the first time it is mentioned in the text.

Acute leukemia is now fully spelled out in line 21

Line 261: rimove the full stop.

Full stop has been removed

Lines 274 and 304: insert the reference.

References are now added

Round 2

Reviewer 3 Report

The authors have revised their original manuscript according to the reviewers’ comments. I think that this revised manuscript is better organized and suitable for publication in International Journal of Molecular Sciences.